# BALANCE, DON'T BOOST: CALIBRATING VISUAL EVIDENCE TO REDUCE OMISSIONS WITHOUT FABRICATIONS

## ABSTRACT

Multimodal Large Language Models (MLLMs) have achieved impressive advances, yet object hallucination remains a persistent challenge. Existing methods, based on the flawed assumption that omission and fabrication hallucinations share a common cause, often reduce omissions only to trigger more fabrications. In this work, we overturn this view by demonstrating that omission hallucinations arise from insufficient confidence when mapping perceived visual features to linguistic expressions, whereas fabrication hallucinations result from spurious associations within the cross-modal representation space due to statistical biases in the training corpus. Building on findings from visual attention intervention experiments, we propose the Visual-Semantic Attention Potential Field, a conceptual framework that reveals how the model constructs visual evidence to infer the presence or absence of objects. Leveraging this insight, we introduce VPFC, a plug-and-play hallucination mitigation method that effectively reduces omission hallucinations without introducing additional fabrication hallucinations. Our findings reveal a critical oversight in current object hallucination research and chart new directions for developing more robust and balanced hallucination mitigation strategies.

## 1 INTRODUCTION

Multimodal Large Language Models (Liu et al., 2023a) (Touvron et al., 2023) (Liu et al., 2024a) have achieved significant advancements in visual-language tasks. Nevertheless, the problem of object hallucination remains unresolved. Object hallucination can be categorized into two types: **omission hallucination**, *where the model fails to identify or describe objects present in the visual input*, and **fabrication hallucination**, *where the model erroneously generates information about objects that do not exist in the input*. Existing studies generally suggest that the causes of both types of hallucination are similar, primarily attributed to over-reliance on statistical bias and unimodal priors.

Under this unified cause hypothesis, current mitigation methods (Leng et al., 2024) typically employ a single strategy to address both omission and fabrication hallucinations simultaneously. However, empirical results indicate that these methods often achieve only limited success in reducing omission hallucinations, and do so at the cost of exacerbating fabrication hallucinations, thereby revealing the limitations of current approaches in understanding the underlying mechanisms. This paper proposes that omission and fabrication hallucinations differ fundamentally in their underlying mechanisms.

Section 3.1 reveals that the cause of **omission hallucinations** lies not only in the limited ability of the visual encoder to recognize fine-grained objects but also in the fact that, even when the MLLM successfully captures the visual features of a specific object during the visual perception phase, the model's confidence in these features remains low during the process of mapping them to linguistic symbols. Therefore, during the generation phase, the model is unable to confidently express the identified objects, leading to omission hallucinations.

In contrast, **Fabrication hallucinations** primarily stem from erroneous associations within the cross-modal joint representation space, as elaborated in Section 3.2. During training, due to the frequent co-occurrence of certain object combinations in large-scale corpora, MLLMs establish overly strong and sometimes unreasonable connections between visual features and semantic concepts. When the visual input contains only a subset of the associated objects, the model, influenced by joint

distribution biases, mistakenly activates descriptions of additional, non-existent objects, leading to fabrication hallucinations.

In Section 3.3, we examine the mapping from visual features to semantic concepts through attention intervention experiments, investigating how the model constructs visual evidence to infer the presence or absence of objects. Building on this analysis, we propose the concept of the Visual-Semantic Attention Potential Field: each visual token is embedded within a potential field, where High-Credibility Visual Regions lie at the bottom of potential valleys, facilitating object confirmation, while Low-Credibility Visual Regions occupy the peaks, making confirmation more difficult and biasing the model toward negation.

Building on the above insights, we introduce a plug-and-play hallucination mitigation method in Section 4, called Visual Potential Field Calibration (VPFC). VPFC operates by recalibrating the confidence assigned to visual evidence during the mapping from visual features to semantic concepts, specifically with respect to object existence. This strategy effectively reduces omission hallucinations while avoiding the introduction of fabrication hallucinations. Extensive experiments on multiple benchmarks, including POPE, MM-Hallucination, CHAIR, and LLaVA-Bench, demonstrate that VPFC achieves State-of-the-Art performance among training-free mitigation approaches. In summary, our contributions are as follows:

- We challenge the common assumption that omission and fabrication hallucinations share the same underlying cause. While existing methods can reduce omission hallucinations, we observe that they often simultaneously exacerbate fabrication hallucinations.
- We conduct an investigation into the distinct mechanisms behind these two types of hallucinations. Our analysis reveals that omission hallucinations stem from insufficient confidence in the mapping of visual features, whereas fabrication hallucinations result from erroneous associations within the cross-modal representation space.
- We introduce the concept of the Visual-Semantic Attention Potential Field, which illustrates how the model constructs visual evidence to infer the presence or absence of objects. Building on this foundation, we propose a plug-and-play hallucination mitigation method, VPFC, which effectively reduces omissions while avoiding the introduction of additional fabrications.

## 2 MOTIVATION: BEYOND THE ASSUMPTION OF UNIFIED HALLUCINATION CAUSES

Object hallucinations fall into two types: **omission hallucination**, where the model misses existing objects in the visual input, and **fabrication hallucination**, where it describes non-existent objects. Current methods for mitigating hallucinations in MLLMs are generally founded on a unified assumption: that both omission hallucinations and fabrication hallucinations stem from the same underlying causes, namely the model's overreliance on statistical biases and unimodal priors during generation. However, this understanding presents clear limitations. In reality, omissions and fabrications may fundamentally differ in their generative mechanisms.

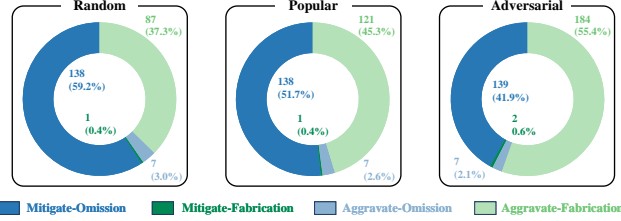

Figure 1: Effects of Visual Contrastive Decoding on the Mitigation and Aggravation of Hallucinations.

Strategies rooted in this unified framework typically seek to address both hallucination types concurrently using the same intervention. For example, Visual Contrastive Decoding (VCD) (Leng et al., 2024) contrasts outputs produced under original versus distorted visual inputs as a corrective mechanism to mitigate the model's excessive dependence on linguistic priors from integrated LLMs

and statistical biases present in pretraining corpora. Nevertheless, in practice, such methods reveal significant shortcomings: while they can partially alleviate omission hallucinations, they often trigger a substantial increase in fabrication hallucinations, thereby further compromising the reliability of model outputs. In the following, we will demonstrate this phenomenon through experiments.

**Experimental Setup.** LLaVA-v1.5-7B served as the backbone MLLM, with greedy search utilized for decoding. We conducted a systematic evaluation of VCD, a well-established method for mitigating hallucinations, analyzing its impact on both the mitigation and exacerbation of omission and fabrication hallucinations. Evaluations were performed using the COCO dataset within the POPE Benchmark (Li et al., 2023c), which focuses on a discriminative task assessing whether the object referenced in a query is present in the visual input.

**Experimental Results and Analysis.** Figure 1 presents the effects of VCD in mitigating and exacerbating two types of hallucinations. While VCD reduced omission hallucinations, it concurrently triggered a notable rise in fabrication ones, particularly on the Adversarial subset, where overall output quality deteriorated. These findings reveal limitations of the unified causality hypothesis.

## 3 ANALYSIS: DIVERGENT ROOTS OF OMISSION AND FABRICATION HALLUCINATIONS

In this section, we systematically investigate the causes of omission and fabrication hallucinations through the use of attention maps and attention intervention. In Section 3.1, we demonstrate that **omission hallucinations** stem from insufficient confidence in mapping perceived visual features to corresponding linguistic expressions. In Section 3.2, we reveal that **fabrication hallucinations** originate from spurious associations within the cross-modal representation space, largely driven by statistical biases in the training corpus.

### 3.1 CAUSE OF OMISSION HALLUCINATIONS

It is widely recognized that a primary cause of omission hallucinations in MLLMs is the limited capacity of their visual encoders, which often struggle with the accurate recognition of fine-grained objects. However, we demonstrate that, in many instances, MLLMs have already encoded effective visual features of the target objects within their latent visual knowledge space, yet fail to articulate this information in the generated textual output.

(Kang et al., 2025) observe that certain attention heads in frozen MLLMs possess strong visual grounding abilities. These heads, which reliably identify object locations relevant to the accompanying text, are referred to as localization heads. Building on this insight, we leverage these localization heads to investigate what visual features are actually captured in the latent visual space of MLLMs when omission hallucinations occur.

Figure 2 illustrates a representative case of an omission hallucination. In the visual input, a person is holding a spoon. However, when prompted with the question "Is there a spoon in the image?", the MLLM produces an omission hallucination by incorrectly responding "no." The prevailing explanation attributes this failure to the small size of the spoon, which supposedly prevents the visual encoder from capturing its features. Contrary to this view, attention maps from the model's localization heads reveal that the model did, in fact, attend to the correct region and successfully captured the visual features of the spoon.

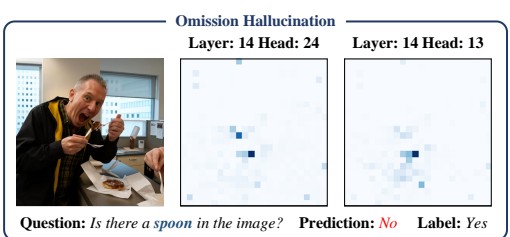

Figure 2: The cause behind omission hallucinations.

These findings suggest that omission hallucinations often do not result from the model's inability to capture meaningful visual features via its visual encoder. Instead, they arise during the mapping from visual representations to semantic concepts, where the model assigns low confidence to the visual evidence. Consequently, the model tends to infer that the object is absent. We provide a more detailed analysis of this mechanism in Section 3.3.

## 3.2 Cause of Fabrication Hallucinations

In contrast to omission hallucinations, fabrication hallucinations occur when the model incorrectly aligns certain visual features with semantic concepts while assigning a high degree of confidence to this misalignment. As illustrated in Figure 3, when presented with an image containing a toilet and asked "Is there a toilet in the image?", the model correctly identifies the visual features of the toilet and maps them to the corresponding semantic concept, yielding an accurate response. However, when asked "Is there a sink in the image?", the model mistakenly interprets part of the toilet's visual features as evidence of a sink, ultimately producing the incorrect answer that a sink is present.

This phenomenon can be attributed to the frequent co-occurrence of sink and toilet within individual training instances in the model's training corpus. As a result, the model may learn to incorrectly align certain visual features of a toilet with the semantic concept of a sink. Consequently, even when the visual input contains only a toilet, the model may infer the presence of a sink based on these overlapping visual cues. This also explains why fabrication hallucinations are particularly prevalent in the Adversarial subset of the POPE Benchmark. In this subset, the queried objects tend to be highly correlated and frequently co-occur in everyday settings. Their visual features and semantic representations are often entangled and misaligned, resulting in more severe cases of fabricated hallucinations.

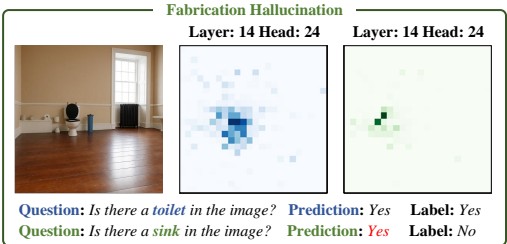

Figure 3: The cause behind fabrication hallucinations.

At a broader level, fabrication hallucinations can be viewed as the result of statistical bias. Yet, current mitigation strategies, designed to correct over-reliance on such biases and unimodal priors, have not effectively reduced these hallucinations. On the contrary, in attempting to mitigate omission hallucinations, they frequently introduce fabrication ones. We explore this mismatch between theoretical motivation and practical results in Section 3.4.

## 3.3 Visual-Semantic Attention Potential Field

In Section 3.1, we demonstrated that omission hallucinations arise when the model correctly captures visual features but assigns low confidence to the corresponding visual evidence. Conversely, in Section 3.2, we showed that fabrication hallucinations occur when the model captures incorrect visual features yet assigns high confidence to them. These findings indicate that the misallocation of confidence plays a central role in the emergence of object hallucinations. This subsection seeks to investigate how the model assigns confidence to visual evidence during the mapping from visual representations to semantic concepts.

We begin by extracting the visual attention maps associated with the model's localization heads. These maps are segmented into two distinct regions: (1) *High-Credibility Visual Regions* (HCVRs), corresponding to areas with high attention scores, and (2) *Low-Credibility Visual Regions* (LCVRs), corresponding to areas with low attention scores. We then apply targeted interventions to each region independently to examine the direct impact of attention manipulation on the recognition performance.

As illustrated in Figure 4, enhancing attention to the HCVRs leads the model to increasingly judge that the queried object is present. In contrast, amplifying attention to the LCVRs causes the model to more frequently conclude that the object is absent. Notably, these effects are consistently observed, regardless of whether the model's initial prediction was correct or whether the object actually appears in the visual input.

These intervention results lead to the following conclusions: (1) HCVRs correspond to areas where visual features have a clear and stable mapping to the semantic concept of the target object. The model consistently interprets these features as positive visual evidence for the presence of the queried object. (2) LCVRs, by contrast, contain visual features that lack a reliable or consistent semantic association with the target object. The model exhibits uncertainty or ambiguity in interpreting these features, effectively treating them as negative visual evidence, indicative of the object's absence.

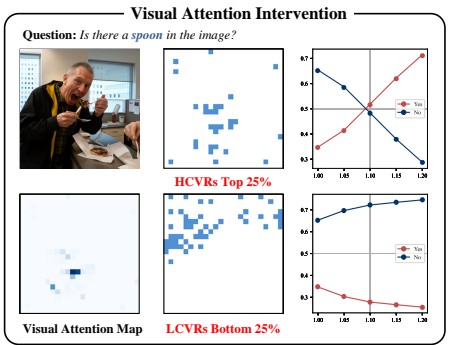
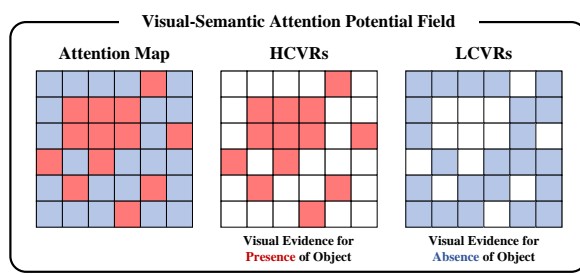

(a) Outcomes of visual attention interventions.     (b) Illustration of the visual potential field.

Figure 4: (a) Visual attention interventions and (b) visual potential field.

When attention to HCVRs is artificially increased, the model receives more salient and reliable visual evidence, thereby boosting its confidence in the presence of the queried object. This attention enhancement effectively activates a high-confidence pathway within the model's visual-to-semantic mapping, reinforcing the alignment between visual features and semantic concepts. In contrast, increasing attention to LCVRs forces the model to extract information from areas that are inherently uncertain or semantically ambiguous. Because the visual-to-semantic mappings in these regions are unstable or unclear, the model is more inclined to draw negative or evasive conclusions, i.e., that the object is absent, as a risk-averse strategy to manage uncertainty.

As shown in Figure 4, we introduce the concept of a Visual-Semantic Attention Potential Field (VSAPF), in which each visual token is embedded within a potential landscape. In this field, HCVRs reside at the bottom of potential wells, zones where the model can readily affirm the presence of an object, while LCVRs are positioned atop potential peaks, where the model encounters greater difficulty in making a positive identification and tends toward negation. The model's reasoning process can be analogized to a ball rolling across the VSAPF: when attention steers the model toward a potential well, it quickly arrives at an affirmative decision; conversely, when attention shifts toward a potential peak, the model is more likely to issue a negative judgment, as a risk-averse response to uncertainty.

### 3.4 OMISSION–FABRICATION IMBALANCE: THE DILEMMA OF CURRENT METHODS

In Section 2, we showed that current hallucination mitigation methods are effective primarily in addressing omission hallucinations. However, while reducing omissions, these methods often exacerbate fabrication hallucinations. Although they are motivated by the goal of correcting the model's over-reliance on statistical biases and unimodal priors, they fail to mitigate fabrication hallucinations that stem from such biases, and in many cases, they inadvertently increase their occurrence. What, then, explains this disconnect between theoretical motivation and empirical outcome?

In Section 3.3, we demonstrated that artificially increasing attention to HCVRs explicitly activates the model's inherent high-confidence pathways within the visual-semantic mapping. This process amplifies the model's confidence in the visual evidence supporting the presence of an object, regardless of whether the object is actually present. Consequently, if current methods are not genuinely correcting the model's over-reliance on statistical biases and unimodal priors, but are instead merely amplifying attention to HCVRs, thereby reinforcing confidence in object presence, then the observed pattern, mitigating omission hallucinations while simultaneously introducing a large number of fabricated hallucinations, can be fully explained.

To illustrate our point, we take the recently proposed Self-Introspective Decoding (SID) (Huo et al., 2025) as an example to briefly demonstrate that current hallucination mitigation methods are, in essence, equivalent to increasing attention to HCVRs. We consider a MLLM parametrized by $\theta$. The model takes as input a textual query $x$ and a visual input $v$, where $v$ provides contextual visual information to assist the model in generating a relevant response $y$ to the textual query. The response

$y$ is sampled auto-regressively from the probability distribution conditioned on the query $x$ and the visual context $v$. Mathematically, this can be formulated as:

$$y_t \sim p_\theta \left( y_t \mid v, x, y_{<t} \right)$$
$$\propto \exp \operatorname{logit}_\theta \left( y_t \mid v, x, y_{<t} \right) \tag{1}$$

where $y_t$ denotes the token at time step $t$, and $y_{<t}$ represents the sequence of generated tokens up to the time step $(t-1)$.

The core motivation behind SID is to harness the model's introspective capabilities to selectively retain visual information by adaptively evaluating the importance of visual tokens, with the aim of deliberately amplifying and suppressing specific vision-text association hallucinations. To this end, SID modifies the model architecture by preserving only a small subset of image tokens with low attention scores after the early decoder layers. This adaptive mechanism is designed to encourage the emergence of vision-text hallucinations during auto-regressive decoding. These hallucinations are then intended to be isolated from the original probability distribution, thereby defining a contrastive distribution $p_{\text{sid}}$ as:

$$p_{\text{sid}}(y_i) = \operatorname{softmax}\Big[ \operatorname{logit}_\theta \left( y_i \mid v, x \right) + \alpha \cdot$$
$$\Big( \operatorname{logit}_\theta \left( y_i \mid v, x \right) - \operatorname{logit}_\theta \left( y_i \mid v_{\text{low}}, x \right) \Big) \Big], \tag{2}$$

where $\alpha$ is a tunable hyperparameter controlling the strength of the contrastive adjustment and $v_{\text{low}}$ denotes the low-importance visual tokens.

Correspondingly, we denote the distribution of the predicted outputs after artificially enhancing attention to HCVRs as $p_{\text{enh}}$, defined as:

$$p_{\text{enh}}(y_i) = \operatorname{softmax}\Big[ \operatorname{logit}_\theta \left( y_i \mid v, x \right) + \beta \cdot$$
$$\Big( \operatorname{logit}_\theta \left( y_i \mid v_{\text{high}}, x \right) - \operatorname{logit}_\theta \left( y_i \mid v, x \right) \Big) \Big], \tag{3}$$

where $\beta$ is the hyperparameter that controls the degree of attention enhancement toward HCVRs.

A comparison between Equation 2 and Equation 3 reveals that the two operations are, in essence, dual to each other with respect to their impact on the final decoding outcomes. When $\alpha$ and $\beta$ are appropriately set, the two decoding formulations become effectively equivalent or transformable into one another. Thus, at the decoding level, the methods are mathematically equivalent, the distinction lies only in their computational pathways, not in their underlying semantics.

## 4 PROPOSED METHOD: VISUAL POTENTIAL FIELD CALIBRATION

In the analysis presented in Section 3.3, we identify the following requirements:

- When the object is present, it is essential to enhance HCVRs in order to explicitly activate the high-confidence pathways within the model's visual-semantic connections. This strengthens the model's confidence in the visual evidence supporting the object's presence and helps mitigate omission hallucinations.
- Conversely, when the object is absent, it is necessary to enhance LCVRs, compelling the model to extract cues from uncertain or semantically ambiguous areas. This promotes the generation of negative or avoidant conclusions (i.e., confirming the object's absence), thereby reducing the risk of fabrication hallucinations.

**Focused Region for Visual Potential Calibration.** However, due to the lack of ground truth regarding the presence of the object, we are unable to apply targeted interventions directly. Nonetheless, we observe a consistent pattern: when the object is absent, HCVRs tend to be spatially dispersed, whereas when the object is present, HCVRs are typically more spatially concentrated. Leveraging this observation, we propose the strategy illustrated in Figure 5: (1) First, we compute the centroid of the HCVRs. Specifically, we define HCVRs as the top 25% of visual tokens ranked by attention weights, as this subset generally captures the majority of the target object. (2) Next, we enhance the attention within a concentrated square region centered at the computed centroid. The size of this enhanced region is set to match that of HCVRs.

The advantages of this approach are as follows: (1) When the object truly exists, HCVRs tend to be spatially concentrated, and the region surrounding the centroid typically aligns well with HCVRs. Enhancing attention in this region increases the model's confidence in the visual evidence of the object's presence. As a result, when visual features are mapped to semantic concepts, the model can more confidently infer the existence of the object. (2) When the object is actually absent, HCVRs are generally dispersed, and the region around the centroid often overlaps partially with LCVRs. Enhancing attention in this area thus simultaneously increases the model's confidence in determining that the object is not present. This helps prevent the introduction of new fabrication hallucinations, and may even correct existing ones.

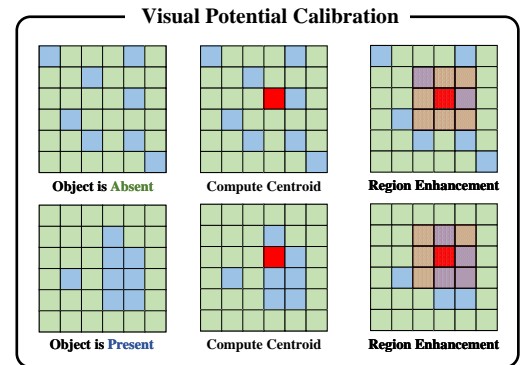

Figure 5: Illustration of visual potential calibration.

**Direct Modification of Hidden States.** While enhancing attention in the centroid region can improve the model's confidence in visual evidence, relying solely on attention adjustment often requires substantial amplification, which may destabilize generation. This is because the model's implicit knowledge is primarily encoded in the hidden states across layers (Burns et al., 2022). To address this, we propose a strategy that computes a confidence-steering direction based on a slight attention boost and directly modifies the hidden states accordingly.

We first apply a mild enhancement (by a factor of 0.05) to the centroid region and compute the difference in hidden states before and after this change to obtain the steering direction $\Delta_{l,h}(x)$:

$$\Delta_{l,h}(x) = h_{l,h}^+(x) - h_{l,h}^-(x), \tag{4}$$

where $h_{l,h}^+(x)$ and $h_{l,h}^-(x)$ represent the hidden states of the $h$-th attention head in the $l$-th layer under the enhanced and original attention conditions, respectively. Next, we apply the following update to the hidden states using a steering coefficient $\alpha$:

$$\tilde{h}_{l,h}(x) = h_{l,h}(x) + \alpha \Delta_{l,h}. \tag{5}$$

This approach enables targeted and effective modification of the model's predictions, while preserving generation stability.

**Selection of Attention Heads.** (Li et al., 2023b) revealed that interventions on hidden states should not be applied across all attention heads, but rather selectively on a subset of the most important ones. Here, we adopt a saliency analysis tool (Michel et al., 2019) to evaluate the importance of all heads. The importance score is computed as:

$$I_{h,l} = \|A_{l,h} \odot \frac{\partial \mathcal{L}(x)}{\partial A_{l,h}}\|_1. \tag{6}$$

where $\mathcal{L}(x)$ denotes the loss function, and $A_{l,h}$ is the attention map of the $h$-th head in the $l$-th layer. Based on the computed importance scores $I_{h,l}$, we select only the top $\gamma\%$ attention heads to perform the intervention.

## 5 EXPERIMENT

Section 5.1 outlines the experimental setup, including the selection of baselines and evaluation tasks. Section 5.2 presents the evaluation results across multiple benchmarks, along with detailed analysis. Section 5.3 reports the results of the ablation studies conducted to assess the proposed method.

### 5.1 EXPERIMENTAL SETUP

**Evaluation Datasets.** To ensure the generalizability of the proposed VPFC method, we evaluated it on a variety of benchmarks encompassing both discriminative tasks (e.g., POPE (Li et al., 2023c)

Table 1: Performance of VPFC on POPE. The best result for each setting is highlighted in bold.

| Model | Method | Random | | Popular | | Adversarial | |
|---|---|---|---|---|---|---|---|
| | | Accuracy | F1-score | Accuracy | F1-score | Accuracy | F1-score |
| LLaVA-1.5 | Regular | 87.10 ↑ 0.00 | 85.53 ↑ 0.00 | 84.83 ↑ 0.00 | 83.33 ↑ 0.00 | 83.60 ↑ 0.00 | 82.29 ↑ 0.00 |
| | VCD | 88.44 ↑ 1.34 | 86.83 ↑ 1.30 | 85.65 ↑ 0.82 | 85.37 ↑ 2.04 | 79.31 ↓ 4.29 | 79.28 ↓ 3.01 |
| | SID | 87.53 ↑ 0.43 | 86.45 ↑ 0.92 | 85.21 ↑ 0.38 | 85.50 ↑ 2.17 | 80.88 ↓ 2.72 | 80.69 ↓ 1.60 |
| | MemVR | 88.50 ↑ 1.40 | 87.34 ↑ 1.81 | 86.10 ↑ 1.27 | 85.01 ↑ 1.68 | 79.20 ↓ 4.40 | 79.28 ↓ 3.01 |
| | **VPFC** | **89.80** ↑ 2.70 | **88.90** ↑ 3.37 | **87.60** ↑ 2.77 | **87.02** ↑ 3.69 | **85.80** ↑ 2.20 | **84.60** ↑ 2.31 |
| Qwen-VL | Regular | 87.43 ↑ 0.00 | 86.48 ↑ 0.00 | 84.70 ↑ 0.00 | 83.96 ↑ 0.00 | 82.50 ↑ 0.00 | 81.70 ↑ 0.00 |
| | VCD | 88.80 ↑ 1.37 | 88.11 ↑ 1.63 | 85.13 ↑ 0.43 | 84.69 ↑ 0.73 | 79.83 ↓ 2.67 | 79.23 ↓ 2.47 |
| | SID | 87.83 ↑ 0.40 | 87.17 ↑ 0.69 | 84.57 ↓ 0.13 | 84.67 ↑ 0.71 | 81.50 ↓ 1.00 | 80.90 ↓ 0.80 |
| | MemVR | 88.47 ↑ 1.04 | 87.62 ↑ 1.14 | 85.27 ↑ 0.57 | 84.73 ↑ 0.77 | 80.90 ↓ 1.60 | 79.80 ↓ 1.90 |
| | **VPFC** | **89.73** ↑ 2.30 | **89.07** ↑ 2.59 | **87.90** ↑ 3.20 | **87.00** ↑ 3.04 | **84.50** ↑ 2.00 | **83.40** ↑ 1.70 |

Table 2: Performance of VPFC on MM-Hallucination. The best result for each setting is highlighted in bold.

| Model | Method | MM-Hall | Object-Level | | Attribute-Level | |
|---|---|---|---|---|---|---|
| | | Total | Existence | Count | Position | Color |
| LLaVA-1.5 | Regular | 620.00 ↑ 0.00 | 185.00 ↑ 0.00 | 146.67 ↑ 0.00 | 128.33 ↑ 0.00 | 160.00 ↑ 0.00 |
| | VCD | 598.36 ↓ 21.64 | 190.00 ↑ 5.00 | 128.33 ↓ 18.34 | 133.33 ↑ 5.00 | 146.70 ↓ 13.30 |
| | SID | 598.33 ↓ 21.67 | 185.00 ↑ 0.00 | 130.00 ↓ 16.67 | 128.33 ↑ 0.00 | 155.00 ↓ 5.00 |
| | MemVR | 610.00 ↓ 10.00 | 190.00 ↑ 5.00 | 130.00 ↓ 16.67 | 130.00 ↑ 1.67 | 160.00 ↑ 0.00 |
| | **VPFC** | **635.00** ↑ 15.00 | **190.00** ↑ 5.00 | **146.67** ↑ 0.00 | **133.33** ↑ 5.00 | **165.00** ↑ 5.00 |
| Qwen-VL | Regular | 618.33 ↑ 0.00 | 175.00 ↑ 0.00 | 140.00 ↑ 0.00 | 128.33 ↑ 0.00 | 175.00 ↑ 0.00 |
| | VCD | 603.33 ↓ 15.00 | 170.00 ↓ 5.00 | 130.00 ↓ 10.00 | 123.33 ↓ 5.00 | 180.00 ↑ 5.00 |
| | SID | 616.66 ↓ 1.67 | 175.00 ↑ 0.00 | 138.33 ↓ 1.67 | 128.33 ↑ 0.00 | 175.00 ↑ 0.00 |
| | MemVR | 608.33 ↓ 10.00 | 170.00 ↓ 5.00 | 135.00 ↓ 5.00 | 133.33 ↑ 5.00 | 170.00 ↓ 5.00 |
| | **VPFC** | **645.00** ↑ 26.67 | **185.00** ↑ 10.00 | **145.00** ↑ 5.00 | **135.00** ↑ 6.67 | **180.00** ↑ 5.00 |

and MME (Fu et al., 2023)) and generative tasks (e.g., CHAIR (Rohrbach et al., 2018) and LLaVA-Bench-in-the-wild (Liu et al., 2023b)). Further details can be found in Appendix B.

**Baseline Selection.** We adopt VCD (Leng et al., 2024), a well-established hallucination mitigation method, alongside two recently introduced State-of-the-Art approaches, SID huo2025selfintrospective and Memory-Space Visual Retracing (MemVR) (Zou et al., 2025), as experimental baselines to facilitate a fair comparison with our proposed method.

**Implementation Details.** We use LLaVA-v1.5-7B (Liu et al., 2024b) and Qwen-VL-7B (Bai et al., 2023) as the MLLM backbones. The enhancement factor, denoted as $\alpha$, is set to 4, and the proportion of selected attention heads, denoted as $\gamma$, is set to $25\%$. Greedy search is used as the decoding strategy in all experiments.

## 5.2 RESULTS AND ANALYSIS

**Results on Discriminative Tasks.** Table 1 presents the experimental results of VPFC on COCO dataset within POPE benchmark. Across the Random and Popular subsets, all methods, including VPFC, exhibit performance improvements. Notably, VPFC demonstrates a more substantial increase in accuracy. We attribute this to VPFC's balanced distribution of confidence between visual evidence indicating the presence and absence of objects. This design helps reduce omissions while simultaneously preventing the introduction of fabrications.

This interpretation is further validated by results on Adversarial subset, where fabrications significantly outnumber omissions (Yin et al., 2025). Existing methods, while somewhat effective in

Table 3: Performance on CHAIR and LLaVA-Bench. Best per column in **bold**.

| Method | CHAIR | | | LLaVA-Bench | | |
|---|---|---|---|---|---|---|
| | CHAIR_S $\downarrow$ | CHAIR_I $\downarrow$ | Average $\downarrow$ | Conversation | Description | Reasoning |
| Regular | 50.2 $\uparrow$ 0.00 | 15.6 $\uparrow$ 0.00 | 32.9 $\uparrow$ 0.00 | 59.6 $\uparrow$ 0.00 | 53.4 $\uparrow$ 0.00 | 75.6 $\uparrow$ 0.00 |
| VCD | 54.8 $\uparrow$ 4.60 | 16.5 $\uparrow$ 0.90 | 35.6 $\uparrow$ 2.70 | 57.4 $\downarrow$ 2.20 | 50.9 $\downarrow$ 2.50 | 76.9 $\uparrow$ 1.30 |
| SID | 49.2 $\downarrow$ 1.00 | 15.1 $\downarrow$ 0.50 | 32.1 $\downarrow$ 0.80 | 59.2 $\downarrow$ 0.40 | 51.3 $\downarrow$ 2.10 | 76.1 $\uparrow$ 0.50 |
| MemVR | 51.2 $\uparrow$ 1.00 | 15.9 $\uparrow$ 0.30 | 33.5 $\uparrow$ 0.60 | 58.1 $\downarrow$ 1.50 | 51.2 $\downarrow$ 2.20 | 77.4 $\uparrow$ 1.80 |
| **VPFC** | **46.8** $\downarrow$ 3.40 | **13.8** $\downarrow$ 1.80 | **30.3** $\downarrow$ 2.60 | **62.1** $\uparrow$ 2.50 | **53.8** $\uparrow$ 0.40 | **77.9** $\uparrow$ 2.30 |

reducing omissions, tend to introduce numerous additional fabrications, thereby degrading overall performance. In contrast, VPFC effectively alleviates omission hallucinations without inducing new fabrications, resulting in improved predictive accuracy even under such conditions.

Table 2 shows performance of VPFC on MME. VPFC maintains or improves accuracy across almost all subsets, whereas existing methods often suffer accuracy drops on certain subsets, highlighting a key issue: their mitigation of omission hallucinations frequently comes at the cost of introducing excessive fabrication errors.

**Results on Generative Tasks.** Table 3 presents the experimental results of VPFC on LLaVA-Bench-in-the-wild, while Table reports results on CHAIR benchmark. Across both generative benchmarks, VPFC consistently outperforms existing methods in prediction accuracy, clearly demonstrating its effectiveness in reducing object hallucinations. Similar to its performance on discriminative tasks, VPFC achieves superior accuracy on generative tasks by effectively mitigating omission hallucinations while avoiding the introduction of additional fabrication hallucinations.

## 5.3 ABLATION STUDIES

We performed an ablation study to investigate the effectiveness of the centroid-focused strategy, using LLaVA-v1.5-7B as the MLLM backbone on the COCO dataset within the POPE benchmark. The study compares different methods for computing the steering direction. Specifically, instead of deriving the confidence steering direction from the concentrated region around the centroid of HCVRs, we compute it directly based on the HCVRs themselves, defined as the top 25% of visual tokens with the highest attention weights.

As illustrated in Figure 6, removing the centroid-focused computation leads to a significant drop in VPFC performance. On the Adversarial subset, the prediction accuracy of VPFC even falls below that of the baseline, reaching the same level as VCD. These results highlight the critical role of the centroid-focused strategy in calibrating the Visual Potential Field. It effectively redistributes confidence across visual evidence regarding object existence, thereby mitigating omissions without introducing additional fabrications. Additional ablation results can be found in Appendix C.

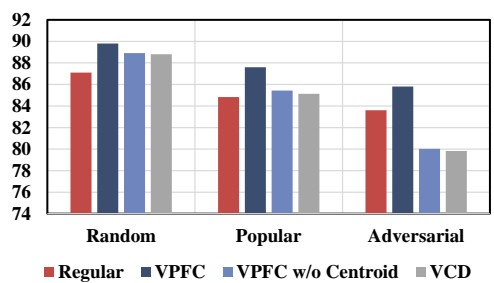

Figure 6: Ablation study on the centroid-focused strategy.

## 6 CONCLUSION

This work challenges the prevailing assumption that omission and fabrication hallucinations share a unified cause, revealing their fundamentally different origins. By introducing VPFC, we demonstrate a training-free approach that effectively mitigates omissions without exacerbating fabrications. Our findings lay the foundation for more balanced hallucination mitigation strategies in MLLMs.

ETHICS STATEMENT

This work aims to mitigate hallucination in multimodal large language models (MLLMs) by improving the reliability of their outputs. Our study does not involve human subjects, personal data, or sensitive demographic information. All experiments are conducted on publicly available benchmarks (e.g., POPE, CHAIR, MME), which are distributed under academic or research-friendly licenses.

By reducing hallucinated or misleading generations, our method has the potential to improve the safety and trustworthiness of MLLMs in downstream applications. Nevertheless, we recognize that efficiency gains and reliability improvements can also accelerate the deployment of large models, including in contexts where risks may arise (e.g., misinformation generation). We stress that such risks stem from downstream misuse rather than from our method itself, and we encourage responsible application of this research in line with the ICLR Code of Ethics.

We affirm that this work complies with ethical research standards, respects dataset usage guidelines, and raises no conflicts of interest or legal concerns.

REPRODUCIBILITY STATEMENT

We have made significant efforts to ensure the reproducibility of our work. Detailed descriptions of our algorithm, experimental protocols, and evaluation metrics are provided in the main paper and appendix. To further support replication, we release runnable code and scripts via supplementary materials. Upon acceptance, we will open-source the complete implementation, including training and evaluation pipelines. All datasets used in our experiments are publicly available, and data processing steps are carefully documented to ensure transparency.

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

## A    RELATED WORK

**Multimodal Large Language Models.** The evolution of MLLMs has progressed from BERT-based decoders to advanced LLM architectures, enabling more effective multimodal relationship modeling. Models such as BLIP-2 (Li et al., 2023a) and MiniGPT-4 (Zhu et al., 2023) employ Q-Former mechanisms to enhance the alignment between visual and textual inputs, facilitating more precise cross-modal interactions. InstructBLIP extends this framework by integrating task-specific instructions, improving the model's ability to interpret context-sensitive visual semantics. Meanwhile,

LLaVA and Qwen-VL adopt simpler linear projection methods that streamline alignment, leading to superior performance in vision-language tasks. Despite these advancements, hallucination remains a persistent challenge that warrants further investigation.

**Hallucination Mitigation Methods.** Visual Contrastive Decoding (VCD) addresses object hallucination by comparing output distributions generated from standard visual inputs and distorted visual inputs. This approach reduces the model's dependence on linguistic priors within integrated LLMs and minimizes the impact of statistical biases in MLLM pretraining corpus. Instruction Contrastive Decoding (ICD) (Wang et al., 2024b), in contrast, focuses on the role of instruction perturbations in amplifying hallucinations. By examining the differences in output distributions between standard and perturbed instructions, ICD detects hallucination-prone content and mitigates its impact effectively.

Building upon these two hallucination mitigation methods, numerous approaches, including Adaptive Focal-Contrast Decoding (HALC) (Chen et al., 2024), Self-Introspective Decoding (SID), and Visual Layer Fusion Contrastive Decoding (VaLiD) (Wang et al., 2024a), have been developed based on similar principles. However, in reality, these methods offer limited relief for omission hallucinations but tend to introduce substantial new fabrications during mitigation.

## B EVALUATION DATASETS

**Polling-based Object Probing Evaluation.** POPE is a novel framework designed to evaluate object hallucinations in MLLMs. Departing from traditional caption-based approaches, POPE frames hallucination detection as a binary task by posing straightforward yes-or-no questions regarding the presence of specific objects in an image (e.g., "Is there a chair in the image?"). Performance on POPE is measured across four metrics: Accuracy, Precision, Recall, and F1 score, allowing for a thorough evaluation of hallucinations in MLLMs.

**Multimodal Model Evaluation.** MME benchmark provides a comprehensive framework for evaluating MLLMs across both perceptual and cognitive dimensions. It consists of ten perception-oriented tasks and four cognition-oriented tasks, with model performance assessed through accuracy metrics. In addition to the full dataset, we leverage specific subsets, such as object existence and counting to analyze object-level hallucinations, while position and color subsets are employed to examine attribute-level hallucinations.

**Caption Hallucination Assessment with Image Relevance.** CHAIR is a metric designed to evaluate how accurately generated captions align with image content. It comprises two components: $CHAIR_i$, which measures object-level hallucinations by calculating the ratio of falsely mentioned objects to all mentioned objects, and $CHAIR_s$, which assesses sentence-level errors by computing the fraction of sentences containing at least one hallucinated object. For evaluation, we use the val2014 split of the MSCOCO dataset, which includes 80 object categories. A random subset of 500 images was selected, and captions were generated using the prompt: "Please describe this image in detail." Together, $CHAIR_i$ and $CHAIR_s$ provide complementary insights into the prevalence and granularity of hallucinated content in image captioning systems.

## C ADDITIONAL ABLATION STUDIES

We performed an ablation study on the attention head selection ratio, using LLaVA-v1.5-7B as the MLLM backbone on the COCO-Random dataset from the POPE benchmark. The objective was to evaluate how different selection ratios impact prediction performance. As illustrated in Figure 7, applying confidence steering intervention across too many attention heads leads to a noticeable decline in prediction accuracy. A more reliable and effective approach is to constrain the selection ratio to $\gamma < 50\%$.

We conducted an ablation study on the steering coefficient, using LLaVA-v1.5-7B as the MLLM backbone on the COCO-Random dataset from the POPE benchmark. The goal was to assess the effect of the steering coefficient on prediction performance. As illustrated in Figure 8, when the coefficient is set within the range $3 < \alpha < 6$, the model consistently yields stable and improved accuracy. These findings suggest that the hyperparameter $\alpha$ possesses a broad and robust tuning range, making it straightforward to configure effectively in practical settings to enhance performance.

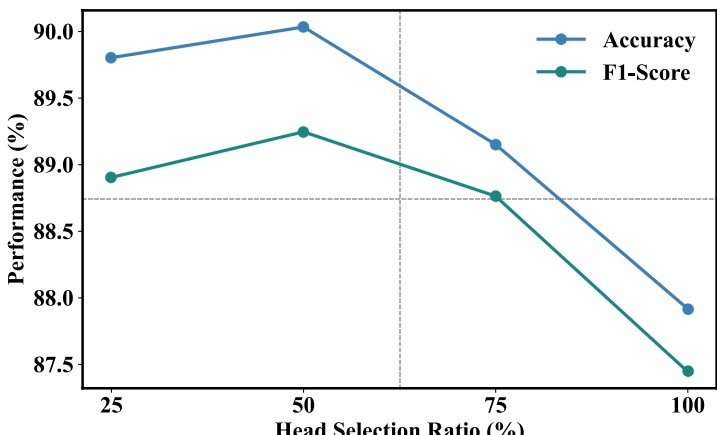

Figure 7: Ablation Study on Head Selection Ratio.

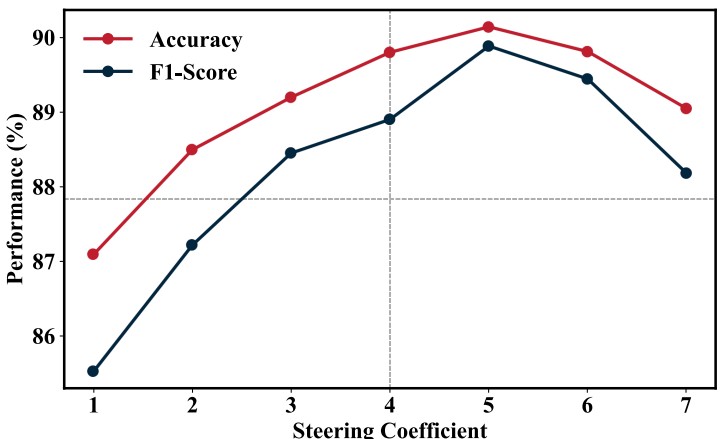

Figure 8: Ablation Study on Steering Coefficient.

# D CASE STUDY ON LLAVA-BENCH

Figure 9 shows a case study of object hallucination mitigation on LLaVA-Bench. It is clear that VPFC effectively mitigates object hallucinations.

# E LIMITATION

While this work provides a detailed analysis of the distinct mechanisms underlying omission and fabrication hallucinations, highlighting that the former arises from low confidence in visual-semantic mapping and the latter from spurious cross-modal associations, our proposed method, VPFC, primarily focuses on mitigating omission hallucinations without inducing fabrication. We do not explicitly target the suppression of fabrication hallucinations. However, this choice does not undermine the method's value: VPFC still achieves state-of-the-art performance among plug-and-play hallucination mitigation approaches, offering the best balance between reducing omissions and avoiding fabrications. Notably, existing training-free methods have consistently failed to suppress fabrication hallucinations, often aggravating them while addressing omissions. Therefore, we believe that identifying the root causes of fabrication hallucinations is a necessary first step, and we leave the development of targeted mitigation strategies as promising future work.

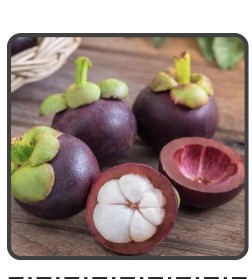

Question: How many uncut fruits are in the image?

**Ground Truth Caption:** 3

**LLaVA-v1.5:** There are five uncut fruits in the image.

**SID:** There are four uncut fruits in the image.

**VPFC:** There are three uncut fruits in the image.

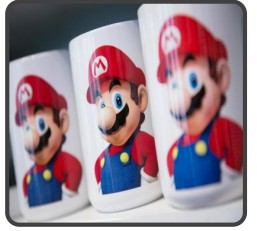

Question: How many coffee mugs are in the set?

**Ground Truth Caption:** 3

**LLaVA-v1.5:** There are four coffee mugs in the set.

**SID:** There are three coffee mugs in the set.

**VPFC:** There are three coffee mugs in the set.

Figure 9: Case Study on Object Hallucination Mitigation on LLaVA-Bench.

