# OpenReview forum: "Balance, Don’t Boost: Calibrating Visual Evidence to Reduce Omissions without Fabrications"
_ICLR.cc/2026/Conference — ICLR 2026 Conference Withdrawn Submission_

### Official Review · Reviewer_Wba2 · 2025-10-29

**Soundness:** 2
**Presentation:** 2
**Contribution:** 2
**Rating:** 2
**Confidence:** 3

**Summary:**

The goal of this work is to mitigate omission hallucination which is defined as a type of hallucination where models fail to identify or describe objects present in the image. Their first contribution is to narrow down the core root cause of omission hallucination to the model's low confidence in mapping the visual features to linguistic symbols. To mitigate the omission hallucinations, this work proposes a plug and play technique called VPFC which recalibrates the confidence assigned to visual evidence during the process of this mapping from visual features to semantic concepts. Concretely, this work partitions attention maps into high-credibility and low-credibility visual regions and applies a centroid-focused adjustment that concentrates the steering signal around the most salient region. Empirically, VPFC improves omission metrics on discriminative tasks (POPE and MME subsets) and on generative benchmarks (CHAIR and LLaVA-Bench) while keeping fabrications in check. The authors report results with LLaVA-v1.5-7B and Qwen-VL-7B using greedy decoding, with ablations showing the centroid step and moderate head selection are important.

**Strengths:**

- The method is training-free and shows improvements on reducing hallucinations across multiple benchmarks such as MM-Hall, POPE, and CHAIR. The effectiveness seems to stand through discriminative as well as generative tasks.
- The paper challenges the common “unified cause” assumption for omissions and fabrications and motivates a different intervention for omissions.

**Weaknesses:**

- Section 3 on systematic analysis of cause of omission and fabrication hallucinations is not very rigorous as the paper only provides only a few illustrative examples. The paper can be made stronger by providing more rigorous evidence for the root causes beyond few illustrative examples to support the hypothesis.
- Some experimental details (exact backbones per table, dataset splits, baseline sourcing) are not always explicit in the narrative. (further raised in questions section)
- Overall the paper is easy to follow, but the readability and organization of the paper can be improved by reducing references of the future sections directly in the Introduction. Typo in lines 453-454 “Table 3 presents the experimental results of VPFC on LLaVA-Benchin-the-wild, while Table reports results on CHAIR benchmark.”.

**Questions:**

- Which dataset is used to evaluate with the POPE method (MSCOCO or something else)?
- Which model is used for evaluation in Table 3?
- For baselines like VCD, SID, and MemVR, which numbers are reproduced versus taken from prior papers, and with what seeds/hyperparameters. For example, the results for MME-Hall did not match with the MemVR’s paper results from a quick look.

---

### Official Review · Reviewer_KB39 · 2025-10-29

**Soundness:** 2
**Presentation:** 2
**Contribution:** 2
**Rating:** 4
**Confidence:** 4

**Summary:**

The paper re-theorizes object hallucination in MLLMs by separating omission (low confidence in mapping visual features to text) from fabrication (spurious cross-modal associations induced by corpus biases). Via visual-attention intervention studies, it proposes a Visual–Semantic Attention Potential Field explaining how models assemble visual evidence for presence/absence judgments. Guided by this framework, the authors introduce VPFC, a plug-and-play mitigation that reduces omission hallucinations without increasing fabrications. Experiments validate that VPFC yields more balanced hallucination control than prior methods premised on a single shared cause, pointing to new directions for robust, evidence-grounded multimodal reasoning.

**Strengths:**

1. The topic is interesting and tries to address an important problem.

2. The authors provide several insightful analyses, which contribute meaningfully to understanding the underlying causes of different hallucination types.

**Weaknesses:**

1. In Figures 2 and 3, please clarify whether “layer” and “head” refer to the **vision encoder** or the **LLM** components.
2. In Figure 4 (right column), specify the **meaning and units of the x- and y-axes** for interpretability.
3. Lines 317–319: the statement about HCVR dispersion vs. concentration is interesting—please include some **quantitative or static analysis** to substantiate this observation.
4. It would help to **emphasize in the main text** that the proposed approach is **training-free**, and to explicitly discuss that while it reduces *omission hallucinations*, it may increase *fabrication hallucinations*, clarifying that this trade-off is intrinsic to training-free methods.
5. Consider adding **stronger baselines** such as **LLaVA-NeXT** and **Qwen2.5-VL** for broader comparison.
6. In the **Related Work** section, include recent studies that mitigate hallucination through **SFT, DPO, or other fine-tuning approaches**.
6. For the benchmark discussion, note that several recent studies [1, 2, 3] address both hallucination and maintain performance (even some improvement) on general scenario. I recommend the authors add some benchmarks like OCRBench, MMMU, MME etc.

[1] Mitigating Object Hallucinations via Sentence-Level Early Intervention.

[2] A topic-level self-correctional approach to mitigate hallucinations in mllms.

[3] Rlaif-v: Aligning mllms through open-source ai feedback for super gpt-4v trustworthiness.

**Questions:**

See above.

---

### Official Review · Reviewer_acjb · 2025-10-31

**Soundness:** 2
**Presentation:** 2
**Contribution:** 2
**Rating:** 2
**Confidence:** 3

**Summary:**

This paper studies object hallucination in Multimodal Large Language Models (MLLMs). It argues that omission and fabrication hallucinations have different causes rather than a shared one. The authors find that omission hallucinations are due to low confidence in mapping visual features to language, while fabrication hallucinations come from spurious correlations in cross-modal representations. Based on this insight, they propose the Visual-Semantic Attention Potential Field (VAPF) framework to explain how models form visual evidence for object presence. Using this framework, they introduce VPFC, a plug-and-play method that reduces omission hallucinations without increasing fabrications. The paper highlights a critical oversight in previous research and offers a new direction for more balanced hallucination mitigation in MLLMs.

**Strengths:**

Strengths
1. The paper gives a clear new perspective by separating omission and fabrication hallucinations and explaining their different causes.
2. The proposed VPFC method is simple, training-free, and shows strong empirical results on multiple benchmarks.
3. The paper is clearly written with good experiments and provides meaningful insights for improving the reliability of multimodal models.

**Weaknesses:**

Weaknesses
1. The VAPF idea remains mostly a qualitative analogy: the paper illustrates “potential wells/peaks” for HCVRs/LCVRs and shows decoding-level equations, but it does not formally define a potential function or prove that attention dynamics follow such a field, so the mechanism risks being descriptive rather than principled.
2. Generalization is uncertain because experiments use only two backbones (LLaVA-1.5-7B and Qwen-VL-7B); results and ablations do not test other popular MLLMs or architectures, making it hard to judge breadth.
3. Some key setup and hyperparameters are under-specified or lightly justified; while α and γ are given and HCVR=top 25% tokens is stated, details like the β setting for attention enhancement and the exact region sizing/rationale are not clearly consolidated, which complicates reproduction and sensitivity assessment.

**Questions:**

1. Can the authors provide a clearer empirical validation of the VAPF concept to show that it genuinely models attention behavior, rather than serving as an interpretive metaphor?
2. How well would VPFC transfer to other multimodal models beyond LLaVA-1.5 and Qwen-VL, and have the authors tried any cross-model generalization experiments?
3. Could the authors clarify how the main hyperparameters (like β, thresholds for HCVR/LCVR, and head selection) were chosen and how sensitive the performance is to these settings?

---

### Official Review · Reviewer_Y2qd · 2025-11-05

**Soundness:** 3
**Presentation:** 2
**Contribution:** 2
**Rating:** 4
**Confidence:** 4

**Summary:**

This paper addresses the problem of object hallucination in MLLMs, focusing on the distinction between omission hallucinations and fabrication hallucinations. The authors challenge the prevailing assumption that both share a common cause and argue instead that omissions arise from low confidence in visual-semantic mapping, while fabrications stem from spurious cross-modal correlations induced by dataset biases. To explore this distinction, the paper introduces the Visual-Semantic Attention Potential Field (VSAPF), an intuitive conceptual model for describing confidence allocation over visual tokens, and proposes Visual Potential Field Calibration (VPFC), a plug-and-play, training-free mitigation approach that recalibrates attention to reduce omissions without causing new fabrications. Experiments on several benchmarks (POPE, MM-Hallucination, CHAIR, LLaVA-Bench) show improvements over baselines such as VCD, SID, and MemVR.

While the paper offers a fresh conceptual angle on multimodal hallucination, it currently lacks theoretical rigor, clear metric interpretation, and sufficient qualitative evidence to substantiate its key claims. The arguments for the distinct origins of omission and fabrication hallucinations remain largely speculative, and the evaluation metrics used do not conclusively demonstrate balanced improvement. Furthermore, the writing could be more concise and better aligned with the standard paper structure.

**Strengths:**

* The work’s main novelty lies in decoupling the underlying causes of omission and fabrication hallucinations, offering a new conceptual perspective. The proposed VSAPF framework provides an intuitive lens for understanding how visual confidence affects object recognition in MLLMs.
* The experimental setup is comprehensive incorporating several benchmarks and ablation studies. The plug-and-play nature of VPFC enhances its practical usability and facilitates potential integration into existing MLLMs.
* Despite some verbosity, the paper presents its ideas in a generally understandable way, with clear figures and visual examples supporting the argument.

**Weaknesses:**

* The claim that omission and fabrication hallucinations originate from distinct mechanisms is not rigorously proven, relying mainly on qualitative attention map analysis and intuitive reasoning. The proposed consistent pattern of focused region (Lines 316-323) and the attention potential field framework lack formal mathematical grounding or verifiable proof.
* Insufficient evidence for empirical claims: The metrics used, accuracy and F1 score, capture aggregate performance but do not independently reflect improvements in omission and fabrication hallucinations. These metrics can increase even if one type of hallucination improves while the other worsens. Thus, the reported gains do not conclusively validate the paper’s core claim of “reducing omissions without fabrications.”
* Only two examples are shown in the appendix, which are insufficient to substantiate the claimed qualitative improvements. A more extensive and systematically analyzed case study section is needed.
* Results are limited to two model backbones (LLaVA and Qwen-VL). Broader testing on more recent or diverse models (e.g., Gemini, InternVL, GPT-4V) would enhance the generality of the findings.
* The sensitivity of enhancement factor and attention head ratio is not deeply analyzed. The reasoning behind chosen values remains unclear.

**Questions:**

* How do accuracy and F1 score separately capture omission versus fabrication hallucinations? Could the authors provide a breakdown (e.g., confusion matrices) showing the relative contribution of each hallucination type?
* Beyond visual inspection, can the authors provide quantitative or theoretical evidence supporting the “potential field” structure in attention maps and the observed spatial concentration pattern (i.e., when the object is absent, HCVRs tend to be spatially dispersed, whereas when the object is present, HCVRs are typically more spatially concentrated)?
* Would VPFC work consistently for larger or instruction-tuned models (e.g., GPT-4V or Gemini)? If not, what limitations prevent its adaptation?
* How much additional inference time or memory overhead does VPFC introduce compared to standard decoding?
* Can the authors include more diverse qualitative examples or visualizations showing concrete cases of omission and fabrication mitigation?
* The method focuses on avoiding new fabrications but does not explicitly address existing fabrication hallucinations. Could the authors outline a potential extension that directly targets fabrication reduction?

---

### Note · Authors · 2025-11-26

I have read and agree with the venue's withdrawal policy on behalf of myself and my co-authors.